# The Use of Calcium Phosphate-Based Starter Pellets for the Preparation of Sprinkle IR MUPS Formulation of Rosuvastatin Calcium

**DOI:** 10.3390/ph16020242

**Published:** 2023-02-06

**Authors:** Krzysztof Cal, Barbara Mikolaszek, Tobias Hess, Markos Papaioannou, Joanna Lenik, Patrycja Ciosek-Skibińska, Helene Wall, Jadwiga Paszkowska, Svitlana Romanova, Grzegorz Garbacz, Daniel Zakowiecki

**Affiliations:** 1Department of Pharmaceutical Technology, Faculty of Pharmacy, Medical University of Gdansk, al. Gen. J. Hallera 107, 80-416 Gdansk, Poland; 2Chemische Fabrik Budenheim KG, Rheinstrasse 27, 55257 Budenheim, Germany; 3Department of Analytical Chemistry, Institute of Chemical Sciences, Faculty of Chemistry, Maria Curie-Skłodowska University, M. Curie-Skłodowska Sq. 3, 20-031 Lublin, Poland; 4Chair of Medical Biotechnology, Warsaw University of Technology, Noakowskiego 3, 00-664 Warsaw, Poland; 5Harke Pharma GmbH, Xantener Straße 1, 45479 Mülheim an der Ruhr, Germany; 6Physiolution Polska sp. z o.o., Skarbowcow 81/7, 53-025 Wroclaw, Poland; 7Department of Pharmacognosy, National University of Pharmacy, Pushkinska 53, 61002 Kharkiv, Ukraine; 8Physiolution GmbH, Walther-Rathenau-Strasse 49a, 17489 Greifswald, Germany

**Keywords:** calcium phosphate-based pellets, sprinkle formulations, MUPS, rosuvastatin calcium

## Abstract

Sprinkle formulations represent an interesting concept of medicinal products aimed at the steadily growing population of patients suffering from swallowing difficulties (dysphagia). In the present work, immediate-release sprinkle MUPS (multiple-unit pellet system) containing rosuvastatin calcium as a model drug substance was successfully developed. The formulation was prepared by drug layering technique using novel calcium phosphate-based starting pellets (PharSQ^®^ Spheres CM) of three different particle sizes. The study showed that the developed multiparticulates were characterized by uniform distribution of coating layers thickness, as well as fast dissolution rate (more than 85% of rosuvastatin calcium dissolved within 30 min, as required by the relevant USP/NF monograph). Rosuvastatin calcium, like other statins, has a bitter, unpleasant taste. Investigations conducted with an electronic tongue suggested that the developed formulation achieved the desired taste-masking efficiency. The effect was found to be particle size-dependent, improving as the size of the multiparticulates increased.

## 1. Introduction

Sprinkle formulations represent a very compelling concept of drug products dedicated to patients with swallowing difficulties (dysphagia), especially suitable for seniors and children. The significance of such a dosage form is indicated by research showing that the percentage of people suffering from dysphagia is increasing worldwide [1]. It has been reported that possibly up to about 40% of American adults are affected by the swallowing problems, which reflects the scale of the problem [2,3].

Sprinkle formulations can be found in the form of sachets, tablets, or hard capsules and may contain powders, granules, or coated pellets, which are intended to be mixed with vehicles such as soft food (e.g., yoghurt, pudding) or juice before administration. This way of taking medicines facilitates swallowing and helps to mask their unpleasant taste, leading to better patient compliance. Furthermore, studies have shown that sprinkle formulations are more acceptable to children than liquid forms such as traditionally used syrups [4,5]. In recent years, numerous types of sprinkled drug products have begun to emerge and are now available in the form of immediate-release (IR), delayed-release (DR), extended-release (XR), or multiparticulate drug delivery systems (MDDS), including multiple-unit pellet systems (MUPS) [6].

Due to the growing relevance of sprinkle preparations, the U.S. Food and Drug Administration (FDA) has issued guidelines that provide recommendations for this dosage form [7,8]. These guidelines specify a maximum particle size limit of no more than 2.5–2.8 mm. The proposed size has been found to ensure that the product can be comfortably swallowed without chewing when administered as a sprinkle. It also emphasized the importance of masking the disagreeable taste or odor of drugs to improve patients’ acceptance of oral dosage forms. Providing such quality attributes to sprinkle formulations is essential to make them attractive to patients in comparison with oral liquid dosage form [5]. It should also be noted that sprinkle formulations are solid dosage forms and have enhanced stability during storage compared to liquids. In addition, they make it possible to give the drug product the required release characteristics (immediate or modified) [4,6]. With this respect, sprinkle formulations in the form of multiparticulate pellet system (multiple-unit pellet system, MUPS) appear to be the most beneficial, as all the critical quality characteristics can be easily achieved by applying the appropriate coating layers [9,10,11].

The research presented here concerned the development of immediate-release (IR) sprinkle MUPS formulations comprising rosuvastatin calcium (RSC) as a model drug substance. RSC multiparticulates were prepared with drug layering technique by depositing it on the surface of neutral starter pellets. These ready-to-use inert beads are spherically shaped starting excipient cores utilized as drug carriers for oral administration. They are produced by agglomeration of commonly used pharmaceutical excipients such as sucrose, microcrystalline cellulose (MCC), isomalt, tartaric acid, and others which have their own pharmacopeial monographs [12]. The study utilized a new type of water-insoluble starter pellets based on anhydrous dibasic calcium phosphate (DCPA), which was recently introduced to the market [9]. RSC is a synthetic drug that belongs to a class of lipid-lowering medicines called statins and controls the production of low-density lipoprotein such as cholesterol. It is commonly used to treat and prevent cardiovascular and coronary heart diseases. This drug is believed to be the most potent statin on the market, and it has the advantage of being water soluble [13,14]. RSC, like other statins, is challenging to formulate into stable dosage form due to its susceptibility to degradation [15,16]. Various strategies have been proposed to overcome this issue, among which the use of inorganic salts of multivalent metals as stabilizing agents seems to be the most popular. The presence of multivalent inorganic cations suppresses the susceptibility of RSC to oxidative degradation and lactonization [17,18,19,20]. The stabilizing effect of calcium phosphates on the chemical stability of RSC has been described elsewhere [21], and in this respect the use of a starter pellet based on anhydrous calcium phosphate appears to be particularly beneficial. 

DCPA pellets are a novel product and a unique solution offered on the market. These pellets have elevated bulk and tapped density (around 1 g/cm^3^), resulting from very high content of anhydrous dibasic calcium phosphate (80% *w*/*w*). A combination of two commonly used pharmaceutical excipients, brittle calcium phosphate with 20% *w*/*w* of plastic material, microcrystalline cellulose, provides sufficient mechanical strength, including very low friability. Limited friability of starter pellets is vital because it prevents the undesired dust formation, especially at the beginning of the coating process. A summary of DCPA pellets characteristics in comparison with other starter pellets can be found elsewhere [9,22,23].

The aim of the research was to develop IR MUPS containing RSC based on novel DCPA starter pellets. Like other statins, RSC is reported to cause an unappealing taste in the mouth and even affect the sense of taste [24,25]. Therefore, one of the technological challenges was to mask the taste that some patients might find intolerable. The taste-masking effectiveness of the developed formulations was tested with the help of an electronic tongue, which has already been described in detail elsewhere [26,27]. The study also investigated the effect of inert core particle size on the dissolution rate of the model drug substance in different types of dissolution media, as well as in soft food products or liquids that can be used as vehicles for sprinkle drug administration. Considering that the formulations developed here were IR, the latter aspect appears to be very important regarding the taste of RSC, which, if excessively released into the vehicle, can negatively affect the patient compliance.

## 2. Results

### 2.1. Characterization of RSC Multiparticulates

The samples of the developed RSC multiparticulates of each size (S, M or L) were examined with a SEM microscope as per description in Section 4.1. The SEM microphotographs in Figure 1 show the general overview of the RSC multiparticulates, as well as a close-up on the details of their coating. The individual coating layers are marked in the figure with numbers 1 and 2. Figure 2 compares the thickness of the individual coating layers compiled with the surface area and particle size (median circle equivalent (CE) diameter) of RSC multiparticulates measured with an optical particle analyzer as described in Section 4.1.

### 2.2. Effectiveness of Taste Masking

The samples of the developed RSC multiparticulates of each size (S, M, or L) and RSC itself were tested according to the procedure described in Section 4.2. The similarity of the tested samples was evaluated applying the PCA method, and the resulting PCA score plot is shown in Figure 3.

Each of the sample types tested formed a separate, distinct cluster. The clusters of RSC multiparticulates of different size (S, M, and L) were easily distinguishable from the cluster of model drug substance. The highest similarity was observed between M and L sizes, due to their similar value on the first principal component, capturing most of the variance of the dataset (78%). Their clusters were markedly distant from the cluster of the model drug substance. RSC multiparticulates of S size also formed a separate cluster, but its PC1 value was close to that of RSC, and the PC2 value distinguishes them very clearly, although it captures a smaller variance of about 8%. The dissimilarity of RSC multiparticulates to the medicinal substance can be ranked as follows: M ≈ L > S, which suggests quite effective and similar masking of the taste in the M and L size multiparticulates, and slightly less effective masking in the case of S size multiparticulates.

### 2.3. Dissolution Test

The dissolution test was carried out as described in Section 4.3 in two media, i.e., 0.1 M hydrochloric acid and 0.05 M sodium citrate buffer pH 6.6. A comparison of the results obtained for the 5 mg dose is shown in Figure 4 and for the 40 mg dose in Figure 5. The solid lines indicate the release curves recorded in the medium according to the USP/NF monograph for Rosuvastatin Tablets, and the dashed line those in 0.1 M HCl. The red dotted line marks the requirement set in the pharmacopoeial monograph, i.e., no less than 85% of the drug is released within 30 min of the test.

### 2.4. Release of RSC on Various Vehicles

Tests showing how much of the model drug was released from the RSC microparticulates when they come into contact with soft food or juice used as carriers for drug delivery were conducted as described in Section 4.4. Figure 6 displays the results of analyses performed for apple juice, applesauce, kefir, and rice gruel. The target RSC strength and the size of the starter pellets used to make the respective multiparticulates is given in brackets next to the dose tested. A comparison of the chemical stability of RSC in contact with different vehicles is shown in Figure 7. Each column represents the sum (total) of impurities found in the samples expressed as a percentage and calculated by area normalization method.

## 3. Discussion

There are currently only a few RSC formulations presented on the market in the form of IR tablets in doses of 5, 10, 20, and 40 mg. However, such dosage form can be problematic for elderly patients suffering from swallowing difficulties [28,29,30]. To address this matter, single RSC sprinkle formulations in the form of capsules comprising extended-release granules have been commercialized relatively recently. The capsules can be opened, and the contents ingested together with soft food or juice [31,32].

In the presented studies, IR sprinkle MUPS formulations containing RSC as a model drug substance were developed. They were prepared by depositing the model drug substance onto inert DCPA starter pellets as described in Section 4.1. Such drug-layered pellets (multiparticulates) can be placed in hard capsule shells or sachets and are intended to be sprinkled onto a teaspoon or tablespoon of soft food or liquid and swallowed immediately without chewing. The drug substance is contained in many independent units, which makes the formulation very flexible, so that the dose of the drug can be regulated only by the mass of the RSC multiparticulates applied [33,34,35]. 

Tests carried out with SEM (see Figure 1) revealed that the developed multiparticulates were characterized by a spherical shape with two clearly distinguishable layers of uniform thickness. The inner coating layer (labeled 1) contained RSC, while the outer layer (labeled 2) contained a sweetener, isomalt. Processing of SEM micrographs using specialized software for data visualization and analysis allowed estimation of layer thicknesses. As can be seen in Figure 2, as the diameter of the multiparticulates increased (in sequence: ~480, ~720, and ~865 µm), and the thickness of both coating layers increased (a total of ~24.8, ~41.2, and ~46.9 µm, respectively). The relatively small values of standard deviations suggest a very uniform coverage of the inner cores by both coating layers. The described relationship was related to the decreasing surface area of the pellets (~132, ~83, and ~56 cm^2^/g, respectively). Although an increase in layer thickness with growing particle sizes was noted, in relation to their diameter, the coating was very similar and stood at 5.2, 5.7, and 5.4%, respectively.

Studies of the effectiveness of taste masking performed with an electronic tongue suggested that in all the developed formulations the desired effect of covering up the bad flavor of the drug was achieved (see Figure 3). Nevertheless, the results showed that the taste masking efficiency depends on the particle size. The most pronounced taste masking capability was determined for RSC multiparticulates of M and L size, and it was slightly worse for these of S particle size. This was probably related to the specific surface area of multiparticulates, which increases with decreasing particle size (compare with Figure 2). In the case of the smallest multiparticulates, the model drug substance was highly exposed to the dissolving medium, which contributed to a faster release of its bitter taste as detected by E-Tongue. The credibility of this type of research employing an electronic tongue has been demonstrated in the published scientific literature [36,37,38]. It is worth noting that although the method gives a good indication of the effectiveness of taste masking, it should not be taken as its definitive confirmation.

The dissolution tests were performed under test conditions as per Test 4 described in USP/NF monograph for Rosuvastatin Tablets in 0.05 M sodium citrate buffer pH 6.6. Since all doses of the product were proportional, the studies were conducted only on extreme doses, i.e., 5 and 40 mg. Under the applied dissolution conditions, the drug substance was released very quickly from all the developed formulations (regardless of their strength or pellet size), so that almost entire dose for the highest and lowest doses was released within the first 15 min of the test (solid lines in Figure 4 and Figure 5). The dissolution profiles of formulations based on DCPA pellets of different sizes (S, M, and L) practically overlap, and no effect of particle size on the rate of dissolution of the RSC could be noticed.

In acidic conditions (pH 1) imitating stomach conditions, RSC dissolves slightly slower than at pH close to neutral (pH 6.6), but still almost complete release of the drug substance occurs within 30 min, as required by the relevant USP/NF monograph (dotted lines in Figure 4 and Figure 5). In the case of preparations with a dose of 5 mg, a slight dependence of the release rate on the size of the core could be spotted, and so the release rate increased as the diameter of the core decreased, as in earlier published research [39,40,41]. Considering the relationship shown in Figure 2, it can be assumed that the rate of release of the drug substance here is influenced by the surface area of the multiparticulates. At the same time, however, it could be noted that the amount of RSC released after 25 min of testing was virtually identical for pellets of all sizes. Such a correlation could not be identified for the 40 mg dose, where the dissolution profiles were almost identical. By comparing the release rates for the 5 mg and 40 mg doses, it could be seen that the release rate was slightly faster from the lower dose formulations. This was probably due to the fact that the mass of the filling for the higher dose was several times bigger than that for the lower one. Thus, in the basket of the dissolution apparatus, the multiparticulates occupied a much larger volume, which hindered the contact of the whole bulk with the dissolution medium. In addition, dissolution of RSC in these conditions may be hindered by the saturation of the solution inside the basket and/or by aggregation of the pellets.

The MUPS formulations developed in this work were IR ones, which can be administered on soft food or juice as a vehicle. However, RSC, like other statins, is characterized by a very disagreeable, bitter taste [42,43]. Therefore, an important issue addressed in this research was the masking of the taste of RSC and the selection of a suitable vehicle in which there would be no excessive release of this bitter drug substance, which in turn could negatively affect the patient’s sensation while ingesting the drug. Masking the unpleasant taste of the RSC was achieved by applying a double coating. In the first phase, the RSC was deposited on an inert core, and in the second phase, a coating based on a polymer (HPMC) and isomalt as a sweetener were employed. 

The study also evaluated how much of the drug substance was released upon contact with such vehicles as soft food (applesauce, kefir, and rice gruel) or liquid (apple juice) as described in Section 4.4. The results showed that the choice of vehicle had a substantial impact on the amount of RSC released (Figure 6). The lowest quantity of the drug was dissolved in kefir, whereas the largest in apple juice. Given that both vehicles had an acidic pH, this might be due to their different consistency, water content and chemical composition (e.g., protein-rich kefir was semi-solid, compared to apple juice, which was liquid and consisted mainly of sugars and pectin). In general, the quantity of RSC released from the multiparticulates appeared to increase as the consistency of the vehicle became less dense. A noticeably higher amount of RSC was released from multiparticulates of higher doses. Based on the obtained results, it is difficult to observe an unambiguous effect of the size of the multiparticulates on the quantity of the drug substance dissolved in various vehicles.

The result of chemical stability tests indicate that RSC was stable during prolonged contact with various types of vehicles. Although the drug substance showed a slight chemical degradation in contact with applesauce and gruel when compared to the reference sample (processed without any vehicle), the total impurities level still did not exceed the requirements of the USP/NF monograph for Rosuvastatin Tablets. The lowest level of impurities was recorded for kefir. As a dairy product, kefir is a rich source of calcium. Multivalent calcium cations are known to stabilize statins, including RSC [17,18,19,20,21]. Hence, the stability of the drug substance was the highest in this vehicle. In the case of stability studies, it could be noticed a slight effect of grain size on the chemical stability of RSC, which improved as the multiparticulate size decreased. In this case, it may have been the result of a larger contact area between the layer comprising RSC and the surface of the smaller sized pellets (inert cores). It should be kept in mind that these pellets consist of 80% DCPA [9,22,23], and DCPA consists of about 29.5% *w*/*w* calcium, which can stabilize statins. Summarizing the results of studies with various vehicles, the data suggest that kefir appears to be the most favorable for administration of sprinkle IR MUPS formulation with RSC as the drug substance. On the other hand, the use of beverages such as apple juice could dissolve excessive amounts of the medicinal substance, which would have an unacceptable impact on the taste experience.

## 4. Materials and Methods

Rosuvastatin calcium EP (RSC) from Cadchem Laboratories Limited (Chandigarh, India). Calcium phosphate-based starter pellets (DCPA pellets)—Small size: PharSQ^®^ Spheres CM S, medium size: PharSQ^®^ Spheres CM M, large size: PharSQ^®^ Spheres CM L produced by Chemische Fabrik Budenheim (Budenheim, Germany). Hydroxypropylmethylcellulose (HPMC)—Tylopur^®^ 606 manufactured by ShinEtsu SE Tylose GmbH & Co. KG (Wiesbaden, Germany). Isomalt (E953) from Modecor Italiana s.r.l. (Cuvio, Italy). Talc—Micro ACE P-3 produced by Nippon Talc Co., Ltd. (Osaka, Japan). Polyethylene glycol (PEG 400) from POCH S.A. (Gliwice, Poland). Apple juice Riviva manufactured by Sokpol Sp. z o.o. (Myszkow, Poland) and applesauce Owolovo from Real S.A. (Siedlce, Poland) both with a pH of approximately 3.5. Kefir Robico produced by OSM (Radomsko, Poland) with a pH of around 4.4. Rice gruel manufactured by Nestle Polska S.A. (Warsaw, Poland) with a pH of approximately 7.1.

### 4.1. Preparation and Characterization of RSC Multiparticulates

DCPA pellets were coated in two steps in a ProCepT 4M8-Trix Fluid-bed system (FBS) equipped with a Wurster column (ProCepT nv, Zelzate, Belgium). First, 100 g of starter pellets of each size (i.e., S, M, and L) were coated with a mixture of RSC (10 g), talc (1 g) and HPMC (5 g) suspended in 150 mL of purified water. In the second step, a mixture of HMPC (10 g), isomalt (10 g), and polyethylene glycol (0.5 g) dissolved in 50 mL of water were applied to the drug loaded pellets (multiparticulates). The process parameters used in each step are summarized in Table 1.

The prepared RSC multiparticulates were examined for their morphology, thickness of the coating layers, and uniformity of coating with scanning electron microscopy (SEM) Phenom Pro Generation 5 (Thermo Fisher, Eindhoven, The Netherlands) using an in-line detection mode from 5 to 10 kV, with backscattered (BSD) or secondary electron detector (SED). Cross sections of three randomly chosen multiparticulates of each size were prepared and SEM images were recorded at a magnification of 200 for the general overview, and 2000 magnification for the precise thickness measurement. The open-source software Gwyddion (version 2.62) was used to collect the data and estimate thicknesses of coating layers. Measurements were taken at a minimum of 75 randomly selected locations.

Particle size and surface area of RSC multiparticle were tested with a dynamic image analysis system CAMSIZER^®^ X2 (Retsch Technology GmbH, Haan, Germany). Then, 5 g of each sample was analyzed in triplicate with the help of an X-Fall module.

### 4.2. Analysis of Taste Masking Efficiency with an E-Tongue

The effectiveness of covering up the taste of the model drug substance in the developed RSC multiparticulate formulation has been verified by means of an electronic tongue (E-Tongue). In this work, a sensor array consisted of 16 ion-selective electrodes (two sensors of each type) with plasticized PVC membranes of solid contact architecture was employed. The procedure used for the E-Tongue analysis was a standard measurement protocol previously used to test the taste of various pharmaceutical samples reported in articles published earlier [44,45]. The study used sensors with different selectivity, sensitive to carbonate ions, as well as cation-selective (CAT) and anion-selective (AN) electrodes to increase the multifunctionality of the device. The electrochemical cell can be represented as follows: Ag, AgCl; KCl, saturated│CH3COOLi 1 M│sample solution║membrane║ (solid contact); AgCl, Ag. The electromotive force of the ion-selective electrodes—Orion 90-02 reference electrode—was measured with an Electrochemistry EMF Interface system (Lawson Labs. Inc., Malvern, PA, USA).

Before measurements, the potentiometric sensors were calibrated several times according to a specific procedure to obtain and ensure the correct response of the electrodes. Due to RSC low water solubility (BCS II Class drug) [46,47,48], the calibration curves of the constructed electrodes were tested by measuring the EMF in the concentration range of 2·10^−5^–2·10^−3^ mol/L (for three repetitions). Electrodes with anionic function were sensitive in the above-mentioned concentration range from −45.00 ± 0.50 to −68.83 ± 4.04 mV/decade with correlation coefficient R^2^ from 0.995 to 0.999. For sensors with a cationic function, the Nernst coefficient was in the range from 31.30 ± 2.50 to 45 ± 4.50 with mean R^2^ = 0.993. The sensitivity of carbonate electrode was 43 mV/decade in the range of RSC concentration from 2·10^−4^ to 2·10^−3^ mol/L. 

Measurement procedure of the sensor array was performed for the developed RSC multiparticulates of each size (S, M, or L) and for the model drug substance. In this research, the RSC was employed as the reference standard for the bitter taste. The procedure consisted of the following steps: stabilization of the signal for 5 min (sensors immersed in 50 mL of deionized water), introduction of a sample into the medium, and recording changes in the signal of the electrodes over time. The signal change was registered for 15 min (5 min stabilization, 10 min release) as the change of the potential (Δ EMF) was triggered by the release of the drug substance and excipients from multiparticulates. Five independent determinations were made for each sample tested. Between measurements, the sensors were washed with purified water and dried. The obtained signals were processed applying Principal Component Analysis (PCA) performed in SOLO^®^ software (Eigenvector Research Inc., Wenatchee, WA, USA).

### 4.3. Dissolution Tests

The dissolution test was performed under test conditions as per Test 4 described in USP/NF monograph for Rosuvastatin Tablets, using a basket apparatus PTWS 820D (Pharma Test Apparatebau AG, Hainburg, Germany) equipped with 1000 mL amber glass dissolution vessels. The exact mass of the developed RSC multiparticulates of each size (S, M or L) corresponding to the extreme doses available on the market, i.e., 5 mg and 40 mg, was weighed on a MSE125P-100-DU balance (Sartorius Lab Instruments GmbH & Co. KG, Göttingen, Germany). The samples were placed in baskets, immersed in 900 mL of 0.05 M sodium citrate buffer pH 6.6, and the test was carried out for 30 min at rotational speed of 100 rpm and a temperature of 37 °C. Similarly, the dissolution rate of RSC from the developed formulations was studied under acidic conditions, which mimicked the conditions prevailing in the stomach, in 0.1 M hydrochloric acid. Samples were collected every 5 min, filtered through a Minisart^®^ RC 0.45 µm syringe filter (Sartorius, Goettingen, Germany), and analyzed by a UV/VIS spectrophotometer T70 (PG Instruments Ltd., Leicestershire, UK) at 241 nm. The applied spectroscopic method was initially checked for linearity and specificity as per ICH Q2(R2) guideline (EMEA 1995) and was found to be sufficient for the intended purpose of the analysis. Calibration curves were prepared by making a series of dilutions of the stock solutions (0.5 mg of RSC per mL) in a concentration range of 5–50 µg/mL using either 0.05 M sodium citrate buffer pH 6.6 or 0.1 M hydrochloric acid. The determined R^2^ correlation coefficients amounted to 0.9997 and 0.9996, respectively. An analogous method of RSC analysis has already been described elsewhere [21].

### 4.4. Release of RSC on Various Vehicles

The quantity of RSC released upon contact with different types of vehicles for drug administration was tested for all three multiparticulates sizes (S, M, and L) in amounts corresponding to 5 and 40 mg doses. Apple juice, applesauce, kefir, and rice gruel were tested as possible vehicles for RSC administration. RSC determination was carried out by HPLC method using LC-2050C 3D and LC30AD liquid chromatographs (Shimadzu, Kyoto, Japan) equipped with LiChrospher^®^ 100 RP-18 5 µm LichroCart^®^ 150-4 chromatographic column (MERCK, Darmstadt, Germany) at the wavelength of 242 nm. The mobile phase consisted of 0.15% *v*/*v* aqueous solution of orthophosphoric acid and acetonitrile (1:1 *v*/*v*). The flow rate was maintained at 1.0 mL/min, the column temperature at 40 °C, and the autosampler temperature at 20 °C. The injection volume was set at 40 µL. The analysis time was 5.0 min, and the RSC retention time was approximately 2.6 min.

A calibration curve was prepared by making a series of dilutions of the stock solution (1 mg of RSC per mL) in a concentration range of 4–60 µg/mL. The mobile phase was used as the solvent/diluent. The determined R^2^ correlation coefficient stood at 0.9998. To prepare test solutions, an appropriate mass of multiparticulates of each size was weighed on a Pioneer^®^ PX125D balance (Ohaus Corporation, Parsippany, NJ, USA) directly into 2 mL tubes made of USP Class VI PP (VWR International Sp. z o.o. Gdansk, Poland), and 2 mL of the chosen vehicle was added. The samples were mixed with a Vortex 3 (IKA, Staufen, Germany) for 2 min and then centrifuged (15 min, 24,000 g at room temperature). To obtain the test solution, the clear supernatant was diluted with the mobile phase. All samples were prepared in triplicate. Before analysis, they were filtered through a 0.22 µm PTFE syringe filter (VWR, Leuven, Belgium). 

In addition, as suggested by FDA Guidance for Industry [8], a two-hour chemical stability test was conducted in contact with each vehicle used in this study. For this purpose, a quantity of multiparticulates of each size equivalent to 40 mg of RSC was accurately weighed on a Pioneer^®^ PX125D balance directly into 2 mL tubes made of USP Class VI PP (VWR International Sp. z o.o., Gdansk, Poland), and 2 mL of the chosen vehicle was added. The samples were stirred with a Vortex 3 apparatus for 2 h at ambient temperature. The contents of the tubes were diluted 25 times with the mobile phase and sonicated for 15 min. The resulting samples were analyzed for impurities content using the same HPLC method described above.

For comparison, reference samples of RSC multiparticulates of each size (S, M, and L) that were not mixed with the vehicle were prepared. For this purpose, a mass of pellets corresponding to 40 mg of RSC was sonicated for 15 min in an appropriate volume of mobile phase. All samples were prepared in triplicate and filtered through a 0.22 µm PTFE syringe filter (VWR, Leuven, Belgium) before the analysis. The results were calculated with the area normalization method.

## 5. Conclusions

The study has succeeded in developing a sprinkle IR MUPS formulation of RSC that effectively masked the unappealing taste of the model drug substance, simultaneously maintaining its fast dissolution rate. Studies with vehicles (soft foods and juice) that can be used for sprinkle drug administration have shown the release of a relatively small quantity of RSC, which can be effectively covered up by the taste of the food itself. What should be considered is the consistency of the vehicle. In the case of the developed RSC formulation, foods with thicker consistency proved to be the most beneficial. Furthermore, the additional stabilizing effect of the pellet core consisting of calcium on RSC was confirmed, indicating the key role of proper selection of excipients in the pharmaceutical development.

The research has shown that the developed double-coated formulation allows the preparation of an effective drug product that can be safely used by people with swallowing difficulties, without causing a disagreeable taste sensation in the mouth during administration. Regarding the development of drug products containing substances that may negatively impact the patient’s oral sensation, the use of E-Tongue seems to be an effective tool that gives indications about the effectiveness of masking bitter, unpleasant taste in newly developed formulations.

## Figures and Tables

**Figure 1 pharmaceuticals-16-00242-f001:**
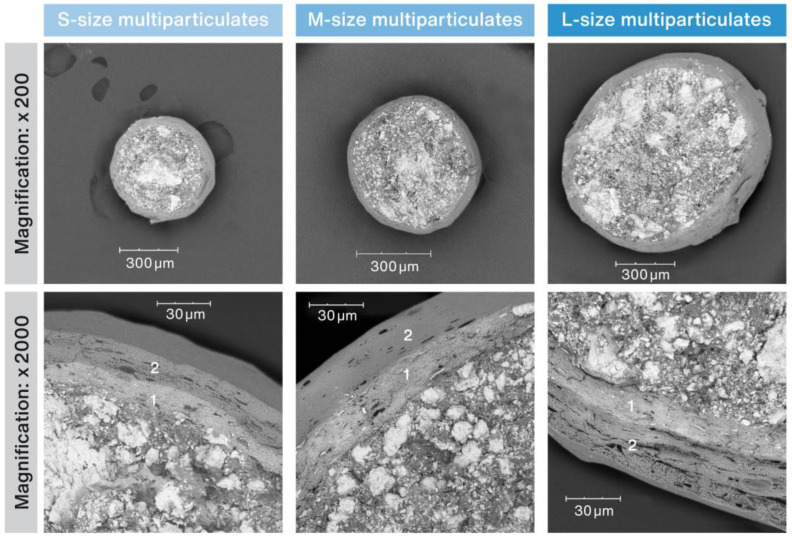
SEM micrographs of cross-sections of RSC multiparticulates (magnification of 200× and 2000×).

**Figure 2 pharmaceuticals-16-00242-f002:**
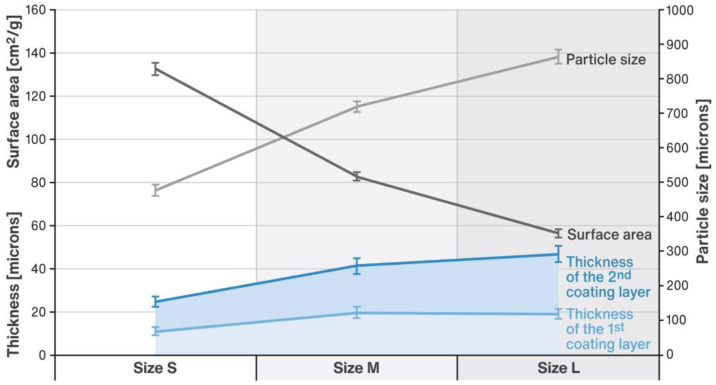
Relationship between RSC multiparticulate size (median CE diameter in µm), surface area (in cm^2^/g), and thickness of the coating layers (in µm); SD is indicated by the error bars.

**Figure 3 pharmaceuticals-16-00242-f003:**
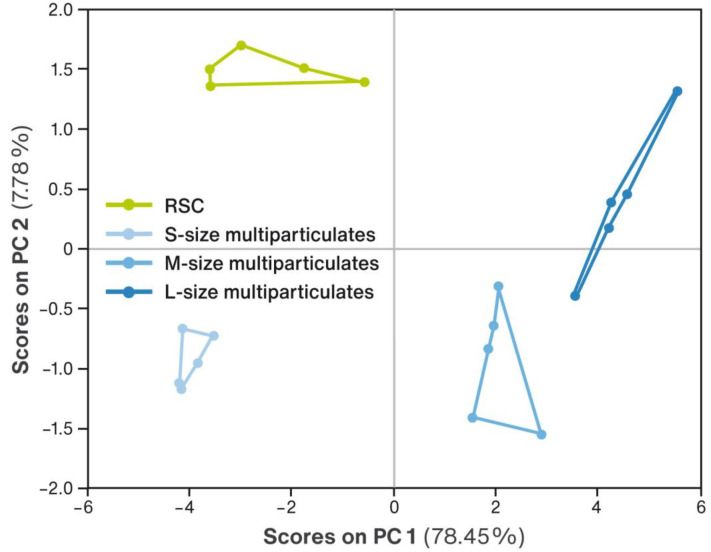
PCA score plot of E-Tongue analysis results for the RSC multiparticulates of each size (S, M, or L), compared to a bitter-tasting reference substance (RSC).

**Figure 4 pharmaceuticals-16-00242-f004:**
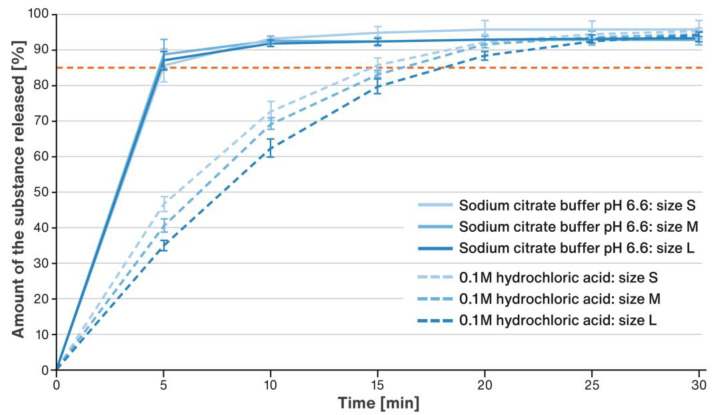
Comparison of the dissolution rate of RSC from a 5 mg sprinkle multiparticulates in 0.05 M sodium citrate buffer pH 6.6 (solid line) and 0.1 M HCl (dotted line); mean values of n = 6, SD is indicated by the error bars.

**Figure 5 pharmaceuticals-16-00242-f005:**
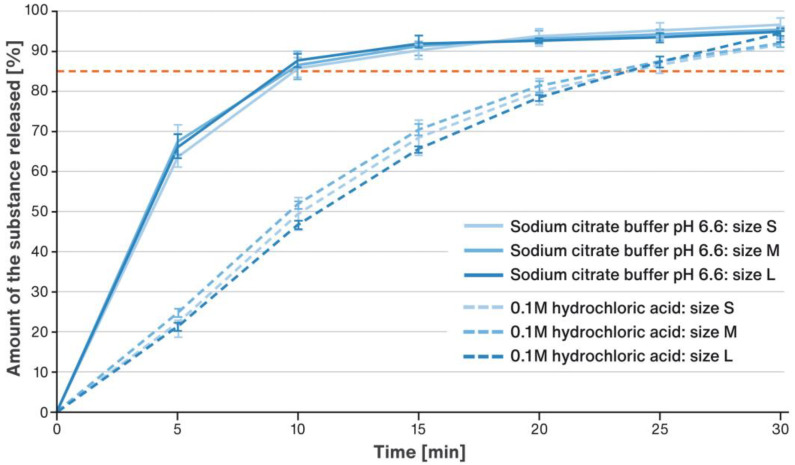
Comparison of the dissolution rate of RSC from a 40 mg sprinkle multiparticulates in 0.05 M sodium citrate buffer pH 6.6 (solid line) and 0.1 M HCl (dotted line); mean values of n = 6, SD is indicated by the error bars.

**Figure 6 pharmaceuticals-16-00242-f006:**
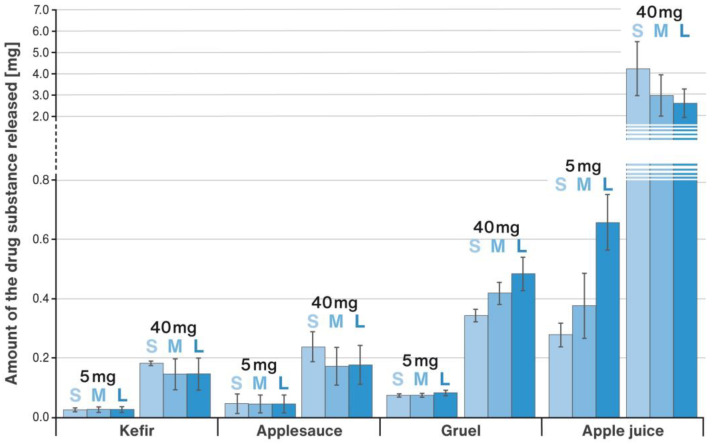
Comparison of the amount of RSC released from S, M, and L size sprinkle multiparticulates during contact with a soft food or liquid; mean values with n = 3, SD is indicated by error bars.

**Figure 7 pharmaceuticals-16-00242-f007:**
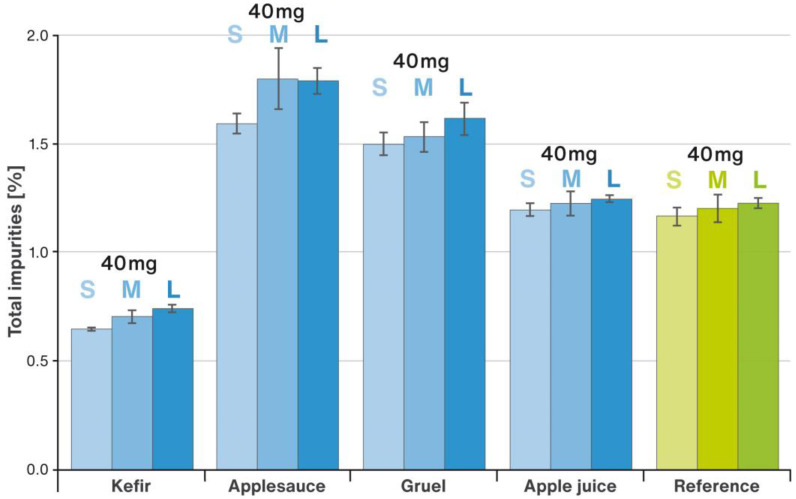
Comparison of the total impurities formed in the S, M, and L size RSC multiparticulates-vehicle mixture with the reference sample; mean values with n = 3; SD is indicated by error bars.

**Table 1 pharmaceuticals-16-00242-t001:** Coating process parameters.

Pellet Size: Coating Step	Inlet Airflow Rate (m^3^/min)	Inlet Air Temperature (°C)	Product Temperature(°C)	Coating Mixture Flow Rate (g/min)	Nozzle Airflow (dm^3^/min)	Spraying Pressure (bar)
Size **S**:step 1	0.19 ± 0.1	65 ± 2	43.5 ± 2	1.1 ± 0.2	8.3 ± 0.1	1.07 ± 0.2
Size **S**:step 2	0.22 ± 0.1	65 ± 2	46.0 ± 2	1.1 ± 0.2	8.7 ± 0.1	1.1 ± 0.2
Size **M**:step 1	0.17 ± 0.1	60 ± 2	42.5 ± 2	1.1 ± 0.2	10.1 ± 0.1	1.07 ± 0.2
Size **M**:step 2	0.19 ± 0.1	65 ± 2	40.0 ± 2	1.1 ± 0.2	10.1 ± 0.1	1.07 ± 0.2
Size **L**:step 1	0.17 ± 0.1	60 ± 2	42.5 ± 2	1.1 ± 0.2	10.1 ± 0.1	1.07 ± 0.2
Size **L**:step 2	0.19 ± 0.1	65 ± 2	40.0 ± 2	1.1 ± 0.2	10.1 ± 0.1	1.07 ± 0.2

## Data Availability

Data is contained within the article.

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
