# Peer review of "The Use of Calcium Phosphate-Based Starter Pellets for the Preparation of Sprinkle IR MUPS Formulation of Rosuvastatin Calcium"

_pharmaceuticals, 2023, doi:10.3390/ph16020242_

Round 1

Reviewer 1 Report

In the manuscript of Cal et al., the authors have studied the dissolution characteristics of sprinkle rosuvastatin pellets based on calcium phosphate-based starter pellets. It is an interesting work well planned although there are some minor questions to be solved by the authors:

1.     The authors have to add a reference to clarify the dissolution specification for IR formulation of rosuvastatin.

2.     What type of drug is rosuvastatin according to the Biopharmaceutics Classification System?

3.     In the line 92 of the introduction, microcrystalline cellulose (MC) is defined as elastic material although MC is usually defined as plastic material. Is MC elastic or plastic?

4.     The total impurities described in figure 7 how were assayed? Possible degradation products are described as impurities?

Author Response

The authors would like to thank the Reviewer for all comments and suggestions, which are addressed below.

  1. The authors have to add a reference to clarify the dissolution specification for IR formulation of rosuvastatin.

As there is currently no pharmacopoeial monograph for such a capsule MUPS formulation for rosuvastatin calcium, the study adopted Test 4 described in USP/NF monograph for Rosuvastatin Tablets. The acceptance criterion used according to this test is: NLT 85% of the drug is released within 30 minutes of the end of the test. Information on the procedure and accepted acceptance criteria is given in the lines 149-153, 220-221, 365-366.

  1. What type of drug is rosuvastatin according to the Biopharmaceutics Classification System?

Rosuvastatin calcium is BCS Class II drug. Information has been added to the manuscript - please check line 340.

  1. In the line 92 of the introduction, microcrystalline cellulose (MC) is defined as elastic material although MC is usually defined as plastic material. Is MC elastic or plastic?

This is an evident mistake that should not happen. Of course, MCC is a plastic material. The mistake has been corrected.

  1. The total impurities described in figure 7 how were assayed? Possible degradation products are described as impurities?

In this work we have used the name "impurities" in accordance with the guideline ICH Q3B (R2) “Impurities in New Drug Product”, in which impurities in new drug products can be classified as degradation products of the drug substance or reaction products of the drug substance with an excipient and/or immediate container closure system.

The impurity content was analyzed using the HPLC method already described in the article, but in the fragment describing the two-hour chemical stability test, there was no reference to this description. This has been corrected in line 418-419.

Reviewer 2 Report

The study covers important topic, it is well conducted and provide interesting and well documented results. I can recommend it for publication in Pharmaceuticals with a suggestion to the authors to perform minor improvements as per following points:

1.       Figure 2 shows properties that may be mutually interconnected as the surface area depends on particle size and the coating thickness at constant amount of material applied is inversely proportional to the surface area. It could be useful to include also numerical data and perform quantitative analysis in paragraph starting at line 194.

2.       The taste masking observation indicates the dissimilarity between the bitter tasting drug and the coated systems. I think that dissimilarity alone is not sufficient to claim good masking efficiency. I agree it is a good indication, but unless verified it cannot be ultimately claimed.

3.       The dissolution profiles show marked sensitivity of release rate between 5 and 40 mg tests. While the release rate seems to be controlled by surface area of the multiparticulates for 5 mg doses, the differences among S, M and L are minimal for 40 mg doses. The release mechanism may be controlled by saturation within the basket or by aggregation of the pellets. Although some hints of discussion is present around line 234, I recommend deeper discussion of measured data.

Author Response

The authors would like to thank the Reviewer for all comments and suggestions, which are addressed below.

  1. Figure 2 shows properties that may be mutually interconnected as the surface area depends on particle size and the coating thickness at constant amount of material applied is inversely proportional to the surface area. It could be useful to include also numerical data and perform quantitative analysis in paragraph starting at line 194.

According to the Reviewer's recommendation, relevant information was added in lines 199 – 206.

  1. The taste masking observation indicates the dissimilarity between the bitter tasting drug and the coated systems. I think that dissimilarity alone is not sufficient to claim good masking efficiency. I agree it is a good indication, but unless verified it cannot be ultimately claimed.

The reviewer's remark is absolutely correct. Therefore, the article did not use categorical statements. Rather, it said that the data suggested the effectiveness of taste masking (please see lines 207-208), or that they were a good indication of the effectiveness of taste masking (check line 440-443. This matter has been additionally clarified in lines 216-219.

  1. The dissolution profiles show marked sensitivity of release rate between 5 and 40 mg tests. While the release rate seems to be controlled by surface area of the multiparticulates for 5 mg doses, the differences among S, M and L are minimal for 40 mg doses. The release mechanism may be controlled by saturation within the basket or by aggregation of the pellets. Although some hints of discussion is present around line 234, I recommend deeper discussion of measured data.

Suggested relevant comments were added to the Discussion in lines 246-248.

Reviewer 3 Report

The use of calcium phosphate-based starter pellets for the preparation of sprinkle IR MUPS formulation of rosuvastatin calcium

The submitted manuscript is dealing with the formulation of immediate-release sprinkle MUPS (Multiple Unit Pellet System) containing rosuvastatin calcium as a model drug substance. The research includes the in vitro characterization of the sprinkle multiparticulate system: SEM, particle size and surface area study, analysis of taste masking efficiency with an E-Tongue, dissolution, release of rosuvastatin on various vehicles and a two-hour chemical stability test was conducted in contact with each vehicle used. The results presented are interesting and support the potential of the formulation approach applied. However, paper needs corrections prior to publication. There is no statistical analysis described in text, moreover, there should be a separate section be included in the manuscript that describes methods and software's used to carried out statistical analysis. This manuscript is written in good English.

Major comments

- Please, include a statistical analysis for comparing results from different pellets size and dose of MUPS, moreover, include the method used for carrying out this analysis.

Minor comments

-References. Kindly elaborate more on all sub-sections of the Discussion Section with references. Appropriate references are required to support the claim and results of any study.

- Please include in Dissolution tests at Material and Methods section, the wavelengths, the concentration ranges and the calibration curves used for UV validation method at both pHs (1.2 and 6.6).

- - Please include in Release of RSC on various vehicles at Material and Methods section, the wavelength and the calibration curve used for the HPLC method.

-Lines 79-80: “This drug is believed to be the most potent statin on the market, and it has the advantage of being water soluble…” line 325: “Due to RSC low water solubility…”, please clarify how is the solubility of rosuvastatin calcium in order to selection it as a model drug.

Author Response

The authors would like to thank the Reviewer for all comments and suggestions, which are addressed below.
